# Neuron with Steady Response Leads to Better Generalization

**Qiang Fu** [*][†]
Microsoft Research Asia
Beijing, China
qifu@microsoft.com

**Lun Du** [*]
Microsoft Research Asia
Beijing, China
lun.du@microsoft.com

**Haitao Mao** [*][‡]
Michigan State University
Michigan, U.S.
haitaoma@msu.edu

**Xu Chen** [*]
Microsoft Research Asia
Beijing, China
xu.chen@microsoft.com

**Wei Fang** [*][‡]
Tsinghua University
Beijing, China
fw17@mails.tsinghua.edu.cn

**Shi Han**
Microsoft Research Asia
Beijing, China
shihan@microsoft.com

**Dongmei Zhang**
Microsoft Research Asia
Beijing, China
dongmeiz@microsoft.com

## Abstract

Regularization can mitigate the generalization gap between training and inference by introducing inductive bias. Existing works have already proposed various inductive biases from diverse perspectives. However, none of them explores inductive bias from the perspective of class-dependent response distribution of individual neurons. In this paper, we conduct a substantial analysis of the characteristics of such distribution. Based on the analysis results, we articulate the Neuron Steadiness Hypothesis: the neuron with similar responses to instances of the same class leads to better generalization. Accordingly, we propose a new regularization method called Neuron Steadiness Regularization (NSR) to reduce neuron intra-class response variance. Based on the Complexity Measure, we theoretically guarantee the effectiveness of NSR for improving generalization. We conduct extensive experiments on Multilayer Perceptron, Convolutional Neural Networks, and Graph Neural Networks with popular benchmark datasets of diverse domains, which show that our Neuron Steadiness Regularization consistently outperforms the vanilla version of models with significant gain and low additional computational overhead.

## 1   Introduction

Deep Neural Network (DNN) achieves state-of-the-art results in a wide range of areas and has various applications across industries, including self driving cars [37], virtual assistants [38], intelligent healthcare [32], personalized recommendation [34], etc. DNN's success usually relies on a plenty amount of training data. However, DNN's generalization is often hampered in many domains where training data is insufficient because data annotation is labor-intensive and expensive.

---

[*]These authors contributed equally to the work.

[†]Corresponding Author.

[‡]Work performed during the internship at MSRA.

36th Conference on Neural Information Processing Systems (NeurIPS 2022).

Regularization is a popular method that helps to improve generalization through introducing inductive bias. Regularization is one of the key elements of machine learning, particularly of deep learning [8]. Specifically, inductive bias represents assumptions about the model properties other than the consistency of outputs with targets. There have been tremendous efforts in identifying such desired properties, which results in a series of widely used regularization methods. For example, L2 Regularization [36, 21] penalizes large norms of model weights, which puts constraints on "parameter scale". L1 regularization improves "sparseness" by rewarding zero weight or neuron response. Jacobian regularization [42, 13] minimizes the norm of the input-output Jacobian matrix to improve the "smoothness" of the learned mapping function. Orthogonal regularization [4, 1] enlarges "weight diversity" to reduce the feature redundancy. Batch Normalization [15] promotes "training dynamics stability" by reducing the internal covariate shift.

Although the existing works have already proposed various inductive biases from diverse perspectives, including the aforementioned "parameter scale", "sparseness", "smoothness", "weight diversity" and "training dynamics stability", to the best of our knowledge, there is no work to explore inductive bias from the perspective regarding the characteristics of neuron response distribution on each class. From another point of view, the existing works leverage the information related to weights (parameter scale regularization), weight correlations (orthogonal regularization), derivatives of mapping function (smoothness regularization), collective neurons responses (sparseness regularization, Batch Normalization), but none of them considers the intra-class response distribution of individual neurons.

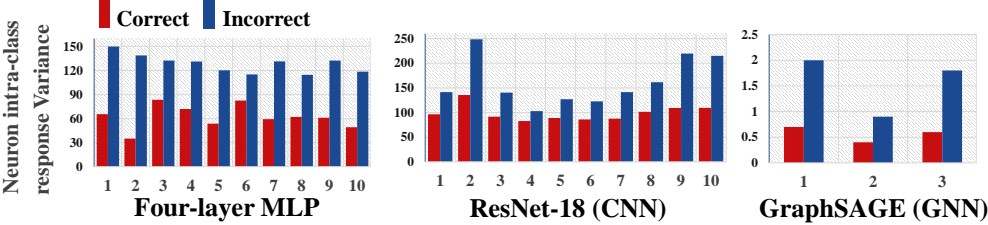

Figure 1: Comparison of the intra-class response variance of correctly and incorrectly classified testing samples for different architectures: four-layer MLP for MNIST, ResNet-18 for CIFAR-10, and GraphSAGE for PubMed. The horizontal axis and the vertical axis represent class indexes and the value of intra-class response variance, respectively. Each bar represents the intra-class response variance aggregated from all neurons in the penultimate layer.

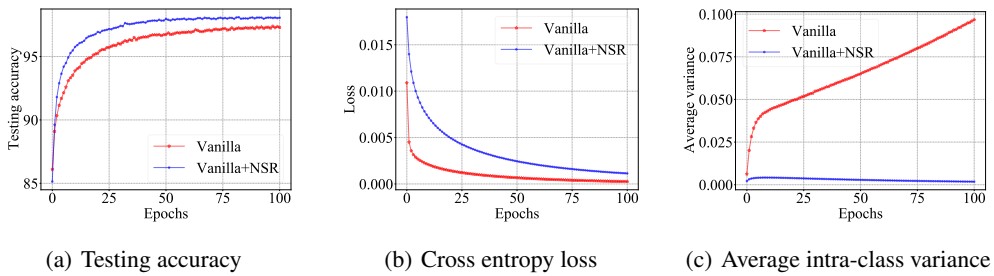

(a) Testing accuracy      (b) Cross entropy loss      (c) Average intra-class variance

Figure 2: Training procedure of the vanilla four-layer MLP and the four-layer MLP with NSR on MNIST. (a) represents the testing accuracy; (b) and (c) illustrate the corresponding cross entropy loss and average intra-class response variance of these two models on the training set.

In this paper, we study the characteristics of the intra-class response distribution of each individual neuron to identify the new regularization method. In more detail, for each individual neuron, we analyze the variance of its response to samples of the same class, which is called **neuron intra-class response variance**. We find that such intra-class response variance has an obvious correlation with classification correctness. As shown in Figure 1, we find the correctly classified samples usually have smaller intra-class response variance compared to the misclassified samples. Besides, it can be observed that the vanilla model with cross entropy as the optimization target usually could not control

intra-class response variance well, as shown in Figure 2, which leaves a potential improvement space for the regularization. The details of experiments and observations are explained in Section 2.

Based on these observations, we articulate a **Neuron Steadiness Hypothesis**: neuron with similar responses to instances of the same class, i.e., smaller neuron intra-class response variance can lead to better generalization. Accordingly, we propose the regularization method called **Neuron Steadiness Regularization (NSR)** to improve generalization by penalizing large neuron intra-class response variance.

Based on the Complexity Measure theory in Section 3.3, we conduct theoretical analysis of the effectiveness of NSR for improving generalization. In addition, our regularization method shows significant improvements in various network architectures, including Graph Neural Networks (GNN), Convolution Neural Networks (CNN), and Multilayer Perceptron (MLP).

To sum up, our contributions are as follows:

- We articulate a Neuron Steadiness Hypothesis and demonstrate its validity. It provides a new regularization perspective based on the neuron-level class-dependent response distribution.

- We propose a new regularization method, Neuron Steadiness Regularization, to improve generalization ability. The method is computationally efficient and general enough to be applied to various architectures and tasks. Theoretical analyses guarantee its effectiveness.

- Extensive experiments are conducted on multiple types of datasets like images, citation graphs and product graphs, with various network architectures including GNN, CNN and MLP. Significant improvements evidently verify the effectiveness of Neuron Steadiness Regularization.

## 2 Observations

In this section, we verify the Neuron Steadiness Hypothesis by experiments with the following identified observations.

### 2.1 Correlation between neuron intra-class response variance and classification correctness

The neuron intra-class response variance is derived from neuron response distribution which could be obtained by recording neuron responses when we feed input samples to the model. In this paper, for the neuron with ReLU as its activation function, we do not take its zero-response into account for calculation, because zero-response corresponds to the inactivated state where the neuron does not respond at all. In other words, for neurons with ReLU as the activation function, we only record its non-zero responses to represent its response distribution.

With the obtained response distribution of any given neuron, it is relatively straightforward to calculate our discussed statistics, i.e., intra-class response variance, which is the mean squared deviation of the neuron response from the mean of the intra-class response. Then, for each neuron, we calculate intra-class response variance corresponding to the correctly classified samples and misclassified samples, respectively. Finally, we aggregate such two respective variances of all neurons in the penultimate layer separately and present them in Figure 1.

Figure 1 shows that for different networks, the average intra-class response variance of correctly classified samples is smaller than that of misclassified ones on arbitrary class. It indicates the strong correlation between classification correctness and neuron intra-class response variance.

### 2.2 Dynamics of neuron intra-class response variance during training procedure

We investigate the tendency of neuron intra-class response variance along with the training procedure. For comparison, we perform the analysis on the vanilla model and the model with our proposed neuron steadiness regularization. We calculate the intra-class response variance of each neuron on the entire training set after each training epoch. Then, the intra-class response variance of all neurons is averaged and denoted as average variance in Figure 2 (c). We also show the testing accuracy and the training cross entropy loss in Figure 2 (a) and (b), respectively. Other architectures demonstrate similar tendencies and can be found in Appendix **??**.

From Figure 2, we could see that the cross entropy objective keeps being optimized during the training procedure which leads to increasing classification accuracy for both models. However, for the vanilla version, the average neuron intra-class response variance is growing larger because the model training does not impose constraints on neuron intra-class response variance. For the model trained with our proposed regularization, the neuron intra-class response variance is controlled and decreases after a few epochs. More importantly, the learned model with well-controlled intra-class response variance has higher testing accuracy than the vanilla version although its corresponding cross entropy loss is even larger. In addition, we could also see that regulating neuron intra-class response variance may also help the optimization procedure to achieve higher accuracy in earlier epochs.

To conclude all the above observations, it is reasonable to design the regularization based on the neuron steadiness hypothesis, i.e., reducing neuron intra-class response variance, for better generalization.

## 3 Method

In this section, we first describe the proposed **Neuron Steadiness Regularization**. Then we further introduce several techniques for computational efficiency.

### 3.1 Definition of Neuron Steadiness Regularization

**Neuron Steadiness Regularization** (NSR) for a specific neuron is defined as the summation of intra-class response variances of different classes. The NSR for the $n$-th neuron can be formulated as:

$$\sigma_n = \sum_{j=1}^{J} \alpha_j \cdot \mathrm{Var}\left(X_{n,j}\right) = \sum_{j=1}^{J} \alpha_j \cdot \mathbb{E}\left[\left(X_{n,j} - \mathbb{E}\left[X_{n,j}\right]\right)^2\right] \tag{1}$$

where $X_{n,j}$ is a random variable denoting the $n$-th neuron's response for a sample belonging to the $j$-th class. $J$ is the number of classes. $\alpha_j = \frac{z_j}{\sum_i z_i}$ is the prior probability of the $j$-th class where $z_j$ is the sample amount of $j$-th class. Notice that $\alpha_j$ is not a hyper-parameter, and it only presents the importance of different classes. With the NSR term defined for each individual neuron, the overall regularization is derived by applying NSR term to all neurons in the network as follows:

$$\mathcal{L}_S = \sum_{n=1}^{N} \lambda_n \sigma_n \tag{2}$$

where $\mathcal{L}_S$ represents the NSR term for the entire network, and $N$ is the number of neurons in the network. $\lambda_n$ is the hyper-parameter to control the regularization intensity. In this paper, for practical simplicity, $\lambda$ is set as the same value for all neurons. Adding the overall regularization term to the main training target, i.e., the cross entropy loss, the final regularized loss function can be written as:

$$\mathcal{L} = \mathcal{L}_C + \mathcal{L}_S = \mathcal{L}_C + \lambda \sum_{n=1}^{N} \sigma_n, \tag{3}$$

where $\mathcal{L}_C$ represents cross entropy loss.

### 3.2 Practical Implementation

#### 3.2.1 Mini-batch Training

In order to use mini-batch training, we adapt our NSR method to do forward and backward propagation based on mini-batches of samples. To be specific, we first transform Eq. (1) as:

$$\begin{aligned}
\sigma_n &= \sum_{j=1}^{J} \alpha_j \left(\mathbb{E}\left[X_{n,j}^2\right] - \mathbb{E}^2\left[X_{n,j}\right]\right) \\
&= \mathbb{E}\left[\sum_{j=1}^{J} \alpha_j X_{n,j}^2\right] - \sum_{j=1}^{J} \alpha_j \mathbb{E}^2\left[X_{n,j}\right].
\end{aligned} \tag{4}$$

The first term has the same form as typical loss functions, i.e., an expectation of the function of data samples, which can be easily generalized to a mini-batch training setting via estimating the expectation with a batch of samples. Although the second term can be estimated in the same way, the square operation magnifies the estimation error, especially when the batch size is not large enough.

To alleviate the problem while not introducing much computing overhead, we propose a memory queue-based estimation method that allows us to leverage more history samples for estimation without additional sampling and forward/backward computation. For each class, we record the number of samples and the summation values of the neurons' response within each batch. To further reduce the storage overhead, we only maintain these values of the latest $M$ batches by two $M$-length queues.

More specifically, an element $c_{m,j}$ in the first memory queue is the count number of $j$-th class instances in the $m$-th batch, represented as:

$$c_{m,j} = \sum_{y_i \in Y_m} \delta\left(y_i = j\right), \tag{5}$$

where $Y_m$ is the set of labels in the $m$-th batch, $y_i$ is the label of $i$-th sample, and $\delta$ is a characteristic function, i.e., $\delta(condition) = 1$ if $condition$ is satisfied, otherwise $\delta(condition) = 0$. An element $s_{m,j}^{(n)}$ in the second memory queue is the summation value of $n$-th neuron's response for the samples belonging to the $j$-th class within the $m$-th batch, represented as:

$$s_{m,j}^{(n)} = \sum_{x_i^{(n)} \in \mathcal{X}_m^{(n)}} \delta\left(y_i = j\right) \cdot x_i^{(n)}, \tag{6}$$

where $\mathcal{X}_m^{(n)}$ is the set of the $n$-th neuron's response within the $m$-th batch and $x_i^{(n)}$ is the $n$-th neuron's response of the $i$-th sample. When a new batch is fed, the estimation of expectation $\mathbb{E}\left[X_{n,j}\right]$ can be updated by the following steps:

$$C_j := C_j - c_{0,j} + c_{*,j}, \quad S_j^{(n)} := S_j^{(n)} - s_{0,j}^{(n)} + s_{*,j}^{(n)}$$
$$\hat{\mathbb{E}}\left[X_{n,j}\right] := S_j^{(n)}/C_j, \tag{7}$$

where $\hat{\mathbb{E}}\left[X_{n,j}\right]$ is the estimation of expectation $\mathbb{E}\left[X_{n,j}\right]$, $C_j = \sum_m c_{m,j}$, $S_j^{(n)} = \sum_m s_{m,j}^{(n)}$, and $c_{*,j}, s_{*,j}^{(n)}$, as new elements appended to the queues, represent the count number and the summation for the new batch. Based on this dynamic update method, the additional memory overhead is negligible, and we give a space complexity analysis in Appendix **??**.

### 3.2.2 Layer Selection criterion for Applying NSR

We find that there is a correlation or redundancy among the steadiness constraints of different layers, meaning that applying NSR on different layers of the network has an overlapping effect on neuron steadiness control. In more detail, from Figure 3, we can see that every layer's variance ratio decreases even only one specific layer is applied with NSR. Such steadiness correlation or redundancy among layers is not surprising because different layers are served as input/output of each other.

Due to the aforementioned redundancy, considering the trade-off between performance gain and computational overhead, we select only one layer to apply NSR. As our NSR is used to reduce neuron intra-class response variance, naturally, the intuitive criterion is to apply NSR to the layer with the largest aggregated neuron intra-class response variance. Detailed experiment results in RQ4 of the experiment section show such a criterion works well. In this paper, if not otherwise specified, we apply NSR to only one particular layer determined by this layer selection criterion.

### 3.3 Theoretical Analysis

In this section, we theoretically analyze the difference in model generalization ability with or without our proposed regularization method NSR based on Complexity Measure [35]. The complexity Measure is one of the mainstream methods to measure the generalization ability of a deep learning model. A **lower complexity measure** means a **better generalization ability**. Formally, a Complexity Measure is a measure function $\mathcal{M} : \{\mathcal{H}, \mathcal{S}\} \rightarrow \mathbb{R}^+$ where $\mathcal{H}$ is a class of models and $\mathcal{S}$ is a training set. According to the definition, if a complexity measure of a given model tends to be 0, the model

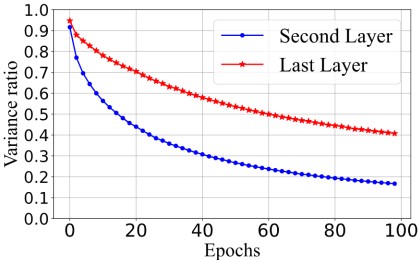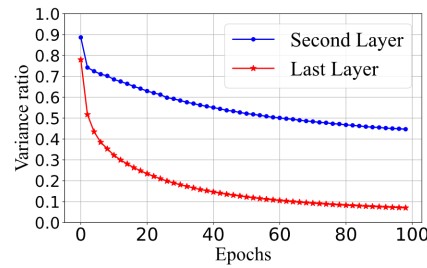

(a) NSR only applied on the second layer    (b) NSR only applied on the last layer

Figure 3: The trend of variance ratio along with the training procedure on MNIST. The variance ratio is the neuron intra-class response variance of the three-layer MLP applied with NSR, (a) only on the second layer and (b) only on the last layer, divided by the corresponding variance of the vanilla three-layer MLP.

should have the best generalization ability. Note that, several different complexity measures have been proposed.

We select Consistency of Representations [33] as the concrete Complexity Measure used in our theoretical analysis. This measure is designed based on Davies-Bouldin Index [5] and is the Winning Solution of the NeurIPS 2020 Competition on Predicting Generalization in Deep Learning. Mathematically, for a given dataset and a model, we define the following statistics:

$$S_i = \left( \frac{1}{n_i} \sum_{\tau}^{n_i} |\mathcal{O}_i^{(\tau)} - \mu_{\mathcal{O}_i}|^p \right)^{1/p} \text{ for } i = 1 \cdots J \tag{8}$$

$$M_{i,j} = ||\mu_{\mathcal{O}_i} - \mu_{\mathcal{O}_j}||_p \qquad \text{for } i, j = 1 \cdots J, \tag{9}$$

where $i$ and $j$ are two different classes, and $n_i$ is the number of samples belonging to the class $i$. $\mathcal{O}_i^{(\tau)}$ is the output representation of the $\tau$-th sample belonging to class $i$ for the final layer, and $\mu_{\mathcal{O}_i}$ is the cluster centroid of the representations of class $i$. $S_i$ is a measure of scatter within representations of class $i$, and $M_{i,j}$ is a measure of separation between representations of classes $i$ and $j$. Then, the complexity measure based on the Davies-Bouldin Index is defined as:

$$\mathcal{C} = \frac{1}{J} \sum_{i=1}^{J} \max_{i \neq j} \frac{S_i + S_j}{M_{i,j}}. \tag{10}$$

Based on the definition of the complexity measure $\mathcal{C}$, we have the following lemma:

**Lemma 3.1.** *For a multi-class classification problem, when (1) a deep learning model utilizing the Cross Entropy loss with an NSR regularization term on any of its intermediate layers is optimized via gradient descent and (2) the capacity of the model is sufficiently large, the Consistency of Representations Complexity Measure on this model $\mathcal{C}$ will tend to be 0. Under the same condition, when the deep learning model is only optimized by the Cross Entropy loss without an NSR regularization term, there will be infinite local minima where the complexity measure $\mathcal{C}$ will be a positive number.*

The proof is shown in Appendix **??**. Intuitively, this lemma shows that in an ideal condition (i.e., the model is sufficiently large), if the definition of the Consistency of Representations Complexity Measure is utilized, a deep learning model with NSR regularization term will be guaranteed to converge to local minima with the best generalization ability. In contrast, the model without NSR will not have the same guarantee.

## 4    Experiment

In this section, we conduct experiments over a variety of datasets to examine the performance of our neuron steadiness regularization on three extensively used neural network architectures. We design a series of experiments to answer the following research questions. **RQ1**: How does NSR perform on various datasets and different neural architectures? **RQ2**: Does NSR outperform other classical

regularization methods? **RQ3**: What is the effect of combining NSR with other popular methods like Batch Normalization or Dropout? **RQ4**: What if we apply NSR to multiple layers instead of one particular layer?

## 4.1 Experiment Setup

**Network Architectures and Datasets**. Three architectures utilized in experiments are Multilayer Perceptron (MLP), Convolutional Neural Network (CNN) and Graph Neural Network (GNN). For vanilla models used as our baselines in different architectures, we adopt ResNet-18 [11], VGG-19 [26] and ResNet-50 for CNN, GCN [17] and GraphSAGE [10] for GNN. We run five MLP models which are denoted as MLP-L, L indicates the number of layers including the input layer. We elaborate on the details of these vanilla models in Appendix **??**. As for benchmark datasets used in our experiments, MLP and CNN are applied to image recognition task on **MNIST** [22], **CIFAR-10** [19] and **ImageNet** [6] datasets, respectively. GNN is applied to node classification on four real-world graph datasets: **WikiCS** [30], **PubMed** [47], **Amazon-Photo** and **Amazon-Computers** [41]. Notice that ImageNet is a large benchmark dataset with 1000 classes. Details are shown in Appendix **??**.

Table 1: Error rate of applying our NSR on five MLP models for MNIST.

| Model | MLP-3 | MLP-4 | MLP-6 | MLP-8 | MLP-10 |
|---|---|---|---|---|---|
| Vanilla (%) | $3.09 \pm 0.10$ | $2.29 \pm 0.07$ | $2.44 \pm 0.09$ | $2.87 \pm 0.09$ | $3.06 \pm 0.06$ |
| Vanilla+NSR (%) | $\mathbf{2.80 \pm 0.08}$ | $\mathbf{1.64 \pm 0.04}$ | $\mathbf{1.76 \pm 0.06}$ | $\mathbf{1.98 \pm 0.09}$ | $\mathbf{1.72 \pm 0.14}$ |
| Gain | 9.39% | 28.38% | 27.87% | 30.87% | 43.79% |

**Experiment Settings**. We follow the typical implementation settings to conduct our experiments. For ResNet-18 and VGG-19 on CIFAR-10, we follow the detailed setting of [48, 9]. For ResNet-50 on ImageNet, we follow the official implementation provided by torchversion library [4]. For GraphSAGE and GCN, we follow the implementation setting of [28]. For MNIST dataset, we divide 60000 training images

Table 2: Error rate of applying our NSR on ResNet-18 and VGG-19 for CIFAR-10, and top-5 error rate on ResNet-50 for ImageNet.

| Model | ResNet-18 | VGG-19 | ResNet-50 |
|---|---|---|---|
| Vanilla | $4.22 \pm 0.07$ | $9.19 \pm 0.18$ | $7.82 \pm 0.07$ |
| Vanilla+NSR | $\mathbf{3.74 \pm 0.08}$ | $\mathbf{8.09 \pm 0.17}$ | $\mathbf{6.98 \pm 0.08}$ |
| Gain | 11.37% | 11.97% | 10.74% |

into the training set with 50000 samples and the validation set with the remaining 10000 samples to select hyper-parameter. For each of the four graph datasets, it is randomly split into training, validation, and testing sets with a ratio of 6:2:2. Note that, SGD [40] is used to optimize MLP, and Adam [16] for other models except for ResNet-18 that is optimized by Momentum [40] according to the implementation setting of [48]. We use the typical setting of batch size as 100 for all experiments. To ensure the model convergence, training epochs are set as 100 for both MLP and GNN, 200 for ResNet and 500 for VGG.

In all experiments except for RQ4, we apply NSR to only one particular layer with the same $\lambda$ for each neuron. For this only one hyper-parameter $\lambda$, like most other regularization methods, we apply a random search strategy to find its proper value ranging from 1e-2 to 10. The error rate is the evaluation metric and each result is averaged over 5 runs with different random seeds. The hardware environments are detailed in Appendix **??**.

## 4.2 Experiment Results

**RQ1: Performance of NSR**: Here, we discuss the performance of NSR over different models and present the results on Tab. 1 - 3 "Gain" is the percentage of relative reduction in the error rate. Tab. 1 demonstrates that NSR can improve the performance of MLP with different layers: the relative error rate is reduced by 9.39% at least and 43.79% at most. Besides, as the number of network layers increases, the gain of NSR shows a roughly upward trend. The four-layer MLP achieves the lowest classification error rate among all vanilla version baselines, and the accuracy becomes worse as the networks grow deeper. It indicates that deep MLP encounters a severe overfitting problem. The

---

[4]https://pytorch.org/hub/pytorch_vision_resnet/

success of our regularization in addressing such a problem reveals the importance of stabilizing the response of each individual neuron to instances from the same class.

For CNN models, Tab. 2 demonstrates that NSR can reduce the relative error rate for CIFAR-10 by 11.97% on VGG-19 and 11.37% on Resnet-18, and reduce the relative error rate (top-5) for ImageNet by 10.74% on ResNet-50. It is worth mentioning that VGG-19, ResNet-18 and ResNet-50 have already adopted Batch Normalization, Dropout regularization and Weight Decay in vanilla models, the accuracy gain of our method reveals that NSR can have extra benefits to ulteriorly enhance generalization ability when combined with Batch Normalization, Dropout and Weight Decay. We will show more evidence about this in RQ3.

Table 3: Error rate of applying our NSR on GCN and GraphSAGE over four graph datasets.

| Dataset | Layers | GraphSAGE (%) | GraphSAGE+NSR (%) | GCN (%) | GCN+NSR (%) |
|---|---|---|---|---|---|
| PubMed | 2 | $10.73 \pm 0.06$ | $\mathbf{9.89 \pm 0.08}$ | $12.02 \pm 0.00$ | $\mathbf{11.92 \pm 0.00}$ |
| | 3 | $10.20 \pm 0.25$ | $\mathbf{9.48 \pm 0.12}$ | $12.76 \pm 0.18$ | $\mathbf{12.19 \pm 0.11}$ |
| | 4 | $10.43 \pm 0.17$ | $\mathbf{9.79 \pm 0.19}$ | $14.01 \pm 0.07$ | $\mathbf{12.96 \pm 0.08}$ |
| Amazon-Photo | 2 | $5.82 \pm 0.00$ | $\mathbf{4.54 \pm 0.10}$ | $6.73 \pm 0.00$ | $\mathbf{6.27 \pm 0.00}$ |
| | 3 | $5.20 \pm 0.14$ | $\mathbf{4.86 \pm 0.13}$ | $8.00 \pm 0.11$ | $\mathbf{7.96 \pm 0.10}$ |
| | 4 | $6.37 \pm 0.30$ | $\mathbf{5.62 \pm 0.59}$ | $10.24 \pm 0.14$ | $\mathbf{9.03 \pm 0.25}$ |
| Amazon-Computers | 2 | $11.37 \pm 0.55$ | $\mathbf{10.47 \pm 0.05}$ | $12.17 \pm 0.07$ | $\mathbf{10.86 \pm 0.03}$ |
| | 3 | $11.88 \pm 1.05$ | $\mathbf{10.22 \pm 0.54}$ | $14.90 \pm 0.25$ | $\mathbf{13.66 \pm 0.12}$ |
| | 4 | $15.49 \pm 0.90$ | $\mathbf{12.86 \pm 0.82}$ | $18.07 \pm 0.74$ | $\mathbf{16.02 \pm 0.23}$ |
| WikiCS | 2 | $16.81 \pm 0.21$ | $\mathbf{16.06 \pm 0.33}$ | $18.41 \pm 0.06$ | $\mathbf{17.99 \pm 0.05}$ |
| | 3 | $15.97 \pm 0.18$ | $\mathbf{15.27 \pm 0.21}$ | $18.66 \pm 0.23$ | $\mathbf{18.10 \pm 0.27}$ |
| | 4 | $16.63 \pm 0.31$ | $\mathbf{15.43 \pm 0.24}$ | $19.21 \pm 0.31$ | $\mathbf{18.84 \pm 0.26}$ |

The number of layers of two GNN models varies from 2 to 4. It is a typical setting following empirical experiences as nodes will capture similar information from neighbors and result in over-smoothness when GNNs grow deeper. Tab. 3 shows that GraphSAGE and GCN applied with NSR outperform the vanilla model with different layer depths on all datasets. Specifically, GraphSAGE and GCN achieve an average improvement of 8.6% and 5.8%, respectively, and 17.0% improvement at most.

**RQ2: Comparison with Other Regularization**: The vanilla ResNet18 here does not use any regularization term and learning rate decay methods for fairness. We first compare our NSR method with several classical regularization methods shown in Tab. 4. Notice that, both our NSR and the other regularization methods for comparison have only one hyper-parameter to tune, and we utilize the same hyper-parameter searching strategy to select the best hyper-parameter values for all of them. As shown in Tab. 4, our NSR performs best among these regularization methods on all three models with different architectures, i.e., MLP-4 (MLP), ResNet-18 (CNN), GraphSAGE (GNN). The results indicate that our NSR has remarkable improvement and is general for various architectures.

**RQ3: Combination with Other Regularization**: We further investigate the effect of combining our NSR with other popular methods like Batch Normalization and Dropout. We conduct the experiments on MLP-4 and the result are organized in Tab. 5. From Tab. 5 we can find that both Batch Normalization and Dropout can reduce error rate compared with vanilla baseline, and adding our NSR on top of them can further promote the performance significantly. It indicates that NSR can provide complementary regularization benefits with Batch Normalization and Dropout.

Table 4: Error rate comparison of different regularization methods on different models.

| | MLP-4 | ResNet-18 | GraphSAGE |
|---|---|---|---|
| Vanilla | $2.29 \pm 0.07$ | $7.96 \pm 0.12$ | $11.37 \pm 0.55$ |
| L1 | $2.27 \pm 0.05$ | $7.83 \pm 0.23$ | $10.81 \pm 0.13$ |
| L2 | $2.27 \pm 0.05$ | $7.67 \pm 0.18$ | $10.68 \pm 0.35$ |
| Jacobian | $2.21 \pm 0.04$ | $7.90 \pm 0.07$ | $11.27 \pm 0.45$ |
| NSR | $\mathbf{1.64 \pm 0.04}$ | $\mathbf{7.20 \pm 0.09}$ | $\mathbf{10.52 \pm 0.22}$ |

Table 5: Combination of our NSR with Batch Normalization (BN) and Dropout (DO) for training.

| MLP-4 | Vanilla | + BN | + BN&NSR |
|---|---|---|---|
| Error rate | $2.29 \pm 0.07$ | $2.22 \pm 0.04$ | $\mathbf{1.62 \pm 0.08}$ |

| MLP-4 | Vanilla | + DO | + DO&NSR |
|---|---|---|---|
| Error rate | $2.29 \pm 0.07$ | $2.19 \pm 0.04$ | $\mathbf{1.64 \pm 0.04}$ |

Table 6: Effect of applying NSR on different layer(s) of MLP-4. The subscript number indicates which layer(s) NSR is applied on.

| | MLP | $\text{MLP}_2$ | $\text{MLP}_3$ | $\text{MLP}_4$ | $\text{MLP}_{3,4}$ |
|---|---|---|---|---|---|
| Error rate | $2.29 \pm 0.07$ | $2.22 \pm 0.08$ | $1.90 \pm 0.13$ | $1.64 \pm 0.04$ | $\mathbf{1.63 \pm 0.08}$ |

Table 7: Effect of applying NSR on different layer(s) of GraphSAGE-2. The subscript number indicates which layer(s) NSR is applied on.

| | GraphSAGE | $\text{GraphSAGE}_1$ | $\text{GraphSAGE}_2$ | $\text{GraphSAGE}_{1,2}$ |
|---|---|---|---|---|
| Error rate | $11.37 \pm 0.55$ | $10.47 \pm 0.05$ | $10.52 \pm 0.22$ | $\mathbf{10.30 \pm 0.17}$ |

**RQ4: Layer selection for applying NSR**: We have introduced how to select one particular layer to apply NSR in section 3.2.2. To evaluate such layer selection criterion, we compare the performances trained by applying NSR to multiple layers or to only one selected layer, respectively. Taking MLP-4 on MNIST and GraphSAGE-2 on Amazon-Computers as two examples, we apply NSR to different layer(s), and their corresponding error rates are listed in Tab. 6 $\sim$ Tab. 7, where $\text{model}_l$ means NSR is applied to the $l_{th}$ layer of the model. Notice that, except the first layer of MLP-4, the input layer, for the second, third, and last layer of MLP-4, their neuron intra-class response variance are 409, 510, and 1660, respectively. For GraphSAGE-2, its neuron intra-class response variance are 4.15 and 2.68, for the first and last layer, respectively. The variance of MLP-4 and GraphSAGE-2 are quite different because of the data characteristic difference between MNIST and Amazon-Computers.

The results in Tab. 6 - 7 show that, no matter which layer(s) is applied with NSR, it could always improve the accuracy compared with the vanilla baseline for both MLP-4 and GraphSAGE-2. In addition, according to our layer selection criterion, applying NSR to the layer with the biggest variance, i.e., the last layer for MLP-4 and the first layer for GraphSAGE-2, could achieve the most significant gain compared with applying NSR to other individual layers. Also, it could achieve similar accuracy compared with the best one obtained by applying NSR to multiple layers. This is the empirical evidence to demonstrate the rationality of our layer selection criterion.

# 5 Related Work

Regularization improves generalization ability by introducing the inductive bias based on prior knowledge. According to the type of prior knowledge, existing works can be roughly categorized into Domain Specific Regularization and Model Generic Regularization.

**Domain Specific Regularization** shows success in various domains including Face Recognition [45, 24, 2, 25], Knowledge Graph Completion [49, 20, 31], Graph Neural Network [3, 14, 39, 46, 27]. In Face Recognition, [2] points out the existence of external factors, such as different situations of environmental illumination, head poses, and facial expressions, which brings the challenge that faces images from the same person may have even larger differences than face images from different persons. To address specific challenges in face recognition, [24] imposes the carefully-designed regularization method which enforces the representations of face instances from the same person to be similar. In Graph Neural Network, the popular model usually suffers from the over-smoothing problem due to the six degrees of separation [18]. [3] statistically analyzes the high correlation between smoothness and the mean average distance (MAD) among node representation. Then a regularization called MADGap is proposed which punishes over smoothness by minimizing MAD.

**Model Generic Regularization** can be categorized into network-wise Regularization, layer-wise Regularization, and neuron-wise Regularization, according to the granularity of the studied properties. Network-wise Regularization encodes the desired property of the entire network, like the sparseness of the network and the smoothness of the mapping function between input and output. L2 Regularization [36, 21] encourages small sum squared magnitude of model weights. [13] introduces an efficient framework to minimize the norm of the input-out Jacobian matrix for noise robustness. [7] encourages model parameters to be uniformly low at some local regions of the loss function. Layer regularization [44, 36] becomes popular due to the success achieved by Batch Normalization [15]. Neuron-wise Regularization is the method with the most fine-grained granularity. Dropout [12, 43] is the typical example belonging to this category. It randomly removes neurons with a certain probability to avoid

strong co-adaption between neurons. Activation Regularization [29] adds the L2 Regularization on masked activations of neurons and Temporal Activation Regularization restricts the difference between RNN outputs at adjacent timesteps. To minimize the variance of sample variance and obtain a few distinct modes, Variance Constancy Loss [23] is applied on all neurons before the activation as regularization. Our proposed Neuron Steadiness Regularization belongs to the neuron-wise regularization that leverages information of individual neuron response distribution.

## 6 Conclusion and Future Work

We explore the inductive bias from the new perspective of class-dependent response distribution of individual neurons. Based on experimental observations, we articulate the Neuron Steadiness Hypothesis and propose the Neuron Steadiness Regularization that penalizes large intra-class neuron response variance. Based on the Complexity Measure, we provide a theoretical analysis of its effectiveness for improving generalization. Additionally, we conduct extensive evaluations on diverse datasets with various network architectures to demonstrate its power. Especially, we demonstrate its effectiveness on the classification task with large models like ResNet-50, and large datasets with many classes like ImageNet with 1000 classes. We systematically consider the border impact, and No risk is found.

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
