# Neuron with Steady Response Leads to Better Generalization

**Qiang Fu** [*] [†]
Microsoft Research Asia
Beijing, China
qifu@microsoft.com

**Lun Du** [*]
Microsoft Research Asia
Beijing, China
lun.du@microsoft.com

**Haitao Mao** [*‡]
Michigan State University
Michigan, U.S.
haitaoma@msu.edu

**Xu Chen** [*]
Microsoft Research Asia
Beijing, China
xu.chen@microsoft.com

**Wei Fang** [*‡]
Tsinghua University
Beijing, China
fw17@mails.tsinghua.edu.cn

**Shi Han**
Microsoft Research Asia
Beijing, China
shihan@microsoft.com

**Dongmei Zhang**
Microsoft Research Asia
Beijing, China
dongmeiz@microsoft.com

## A   Proof

### A.1   Lemma 3.1

*Lemma* 3.1.  For a multi-class classification problem, when (1) a deep learning model utilizing the Cross Entropy loss with an NSR regularization term on any of its intermediate layer is optimized via gradient descent and (2) the capacity of the model is sufficiently large, the Consistency of Representations Complexity Measure on this model $\mathcal{C}$ will tend to be 0. Under the same condition, when the deep learning model is only optimized by the Cross Entropy loss without an NSR regularization term, there will be infinite local minima where the complexity measure $\mathcal{C}$ will be a positive number.

*Proof.*  Let us consider the original Cross Entropy loss first:

$$\mathcal{L}_C = -\sum_{\tau=1}^{\mathcal{N}} \log q(y = y_\tau | x_\tau; \Theta), \tag{1}$$

where $\mathcal{N}$ is the number of training samples, $y_\tau$ is the label of the $\tau_{th}$ sample, $x_\tau$ is the feature of the $\tau_{th}$ sample, $\Theta$ is the parameters of the model, and $q(y|x_\tau; \Theta)$ is the predicted probability modeled by the deep learning model. Considering a single sample, we have the Cross Entropy loss for the $\tau_{th}$ sample:

$$\mathcal{L}_C^{(\tau)} = -\log q(y = y_\tau | x_\tau; \Theta) = -\log q_\tau(y), \tag{2}$$

where $q_\tau(y) \triangleq q(y = y_\tau | x_\tau; \Theta)$ for brevity. Because the deep learning models for the classification task always have a normalization operation (e.g., Softmax) to make the final unconstrained

---

[*]These authors contributed equally to the work.
[†]Corresponding Author.
[‡]Work performed during the internship at MSRA.

36th Conference on Neural Information Processing Systems (NeurIPS 2022).

representation be a probability form, $q_\tau(y)$ has the following form:

$$q_\tau(y) = \frac{a_y^{(\tau)}}{\sum_{j=1}^J a_j^{(\tau)}}, \tag{3}$$

where $J$ is the number of classes, $\mathbf{a}^{(\tau)} = [a_1^{(\tau)}, ..., a_J^{(\tau)}]$ is the final representation for the $\tau$-th sample. The loss can be rewritten as:

$$\mathcal{L}_C^{(\tau)} = -\log \frac{a_y^{(\tau)}}{A_\tau} \tag{4}$$

where $A_\tau = \sum_{j=1}^J a_j^{(\tau)}$ is the normalization factor. Since we will use gradient descent to optimize the objective function, we calculate the partial derivatives of the objective for the representations $h_j^{(\tau)}$:

$$\begin{aligned}
\frac{\partial \mathcal{L}_C^{(\tau)}}{\partial h_y^{(\tau)}} &= \frac{a_y^{(\tau)}}{A_\tau} - 1 \\
\frac{\partial \mathcal{L}_C^{(\tau)}}{\partial h_j^{(\tau)}} &= \frac{a_j^{(\tau)}}{A_\tau}, \text{ for } j : j \neq y.
\end{aligned} \tag{5}$$

Let the gradient be zero, then we obtain the sufficient conditions of the local minima:

$$\begin{aligned}
a_j^{(\tau)} &= 0 \text{ for } j : j \neq y, \\
A_\tau &= a_y, \\
a_y^{(\tau)} &> 0, \\
\sum_j a_j^{(\tau)} &= A_\tau
\end{aligned} \tag{6}$$

In that case, under the assumption that the model is sufficiently large, the converged local minima for the $\tau$-th sample are the vectors where only the $y_\tau$-th entry is an arbitrary positive real number and the other entries are 0.

According to the definition of the Consistency of Representations Complexity Measure Eq. (??), it is easy to see that we have an infinite number of local minima where the measures $\mathcal{S}_i$ and $\mathcal{M}_{i,j}$ for arbitrary classes $i$ and $j$ are finite positive numbers. Thus, the Consistency of Representations Complexity Measure $\mathcal{C}$ will be a positive number in those local minima.

Let us take our proposed NSR regularization term $\mathcal{L}_S$ (refer to Eq. (??)) into consideration. When adding NSR to $k$-th layer, the loss of NSR can be rewritten as:

$$\mathcal{L}_S = \sum_i^J \sum_\tau^{n_i} ||\mathcal{O}_{i,k}^{(\tau)} - \mu_{\mathcal{O}_{i,k}}||^2, \tag{7}$$

where $n_i$ is the number of samples belonging to the class $i$, $\mathcal{O}_{i,k}^{(\tau)}$ is the $k$-th layer output representation of the $\tau$-th sample belonging to class $i$, $\mu_{\mathcal{O}_{i,k}}$ is the cluster centroid of the $k$-th layer representations of class $i$. If we set $\mathcal{O}_{i,k}^{(\tau)}$ as optimization variables, it is easy to see that the global minima is $\mathcal{O}_{i,k}^{(\tau)} = \mu_{\mathcal{O}_{i,k}}$. It means that the intermediate representations of the same class in layer $k$ will be the same vector. Combined with the conditions of the local minima of Cross Entropy loss, we have the optimal final representations of the overall loss, i.e.,

$$\mathbf{a}_i^\tau = [\ \underbrace{0\ldots0}_{i-1 \text{ times}}\ C_i\ \underbrace{0\ldots0}_{J-i \text{ times}}\ ] \text{ for } i = 1 \cdots J, \tag{8}$$

where $\mathbf{a}_i^\tau$ is the final representation vector of the $\tau$-th sample belonging to the class $i$, and $C_i$ is a positive real number that satisfies $C_i \neq C_j$ if $i \neq j$. It means that the final representations for the samples belonging to the same class are the same, while the representations are different for the samples belonging to different classes. It is obvious that in that case, $\mathcal{S}_i$ is zero and $\mathcal{M}_{i,j}$ for arbitrary classes $i$ and $j$ is a finite positive number. Thus, the Consistency of Representations Complexity Measure $\mathcal{C}$ will be 0. The Lemma is proven. □

## B  Hardware and Software Environment

The experiments are performed on two Linux servers (CPU: Intel(R) Xeon(R) CPU E5-2690 v4 @ 2.60GHz, Operation system: Ubuntu 16.04.6 LTS). For GPU resources, two NVIDIA Tesla V100 cards are used for MLP and GNN and ResNet-50 on ImageNet experiments while two Titan V cards are used for other CNN models. The python libraries we use to implement our experiments are PyTorch 1.7.1 and PyG 1.6.3.

## C  Details of Reproduction

### C.1  Open Source Code

We publish our code in Github (i.e., `https://github.com/lundu28/NSR`).

### C.2  Details of Baseline Methods

In this subsection, we detailed the network architectures and the baseline methods used in our experiments.

**MLP** Multilayer Perceptron is a fundamental deep neural network architecture that is composed of an input layer, several hidden layers and an output layer. Five MLP models with different numbers of hidden layers are used to test the performance of neuron steadiness regularization on MNIST for handwritten digit classification. ReLU is used as an activation function in the experiments. We denote MLP-L as an L-layer MLP network. For example, MLP-4 is a 4-layer MLP with 2 hidden layers.

**CNN** Convolutional neural networks are widely used in the computer vision (CV) field. Two famous ones among them, ResNet-18 [3], VGG-19 [7], are chosen as the baseline methods in our paper for the image recognition task on CIFAR-10 data set. ResNet-50 [3] is chosen for the ImageNet dataset [1]. **ResNet-18** [3] is a 20-layer convolutional neural network with residual connections to avoid the problem of vanishing gradient. **ResNet-50** is similar to ResNet-18 except that it is a more expressive 50-layer convolutional neural network. **VGG-19** [10] is another classic deep convolutional neural network architecture. We utilize a variant of VGG-19 adapted for CIFAR-10 [7] classification.

**GNN** Graph neural networks achieve fantastic results on graph structure data. GCN [4] and Graph-SAGE are two classical methods in classifying nodes on real network data. **GCN** [4] is inspired by CNN and introduces spectral graph convolutions as the layer-wise propagation on graphs. **Graph-SAGE** [2] is the first inductive GNN framework that samples neighbor nodes on graphs.

The detailed settings of different architectures of MLP and GNN are shown in Tab. 1. Notice that we only show the architecture of hidden layers. The dimensions of the input and output layer rely on the input feature size and the number of categories of different datasets respectively.

Table 1: The detail architectures of MLP and GNN.

| Model | Hidden layer dimension |
|---|---|
| MLP-3 | [100] |
| MLP-4 | [256, 100] |
| MLP-6 | [256, 128, 64, 32] |
| MLP-8 | [256, 128, 64, 32, 32, 16] |
| MLP-10 | [256, 128, 64, 64, 32, 32, 16, 16] |
| GNN-2 | [100] |
| GNN-3 | [256, 100] |
| GNN-4 | [256, 128, 64] |

Table 2: The final hyperparameter settings $\lambda$ for MLP and CNN

| Model | MLP-3 | MLP-4 | MLP-6 | MLP-8 | MLP-10 | ResNet-18 | VGG-19 | ResNet-50 |
|---|---|---|---|---|---|---|---|---|
| $\lambda$ | 0.025 | 7.487 | 0.054 | 0.254 | 0.416 | 0.05 | 0.05 | 0.40 |

Table 3: The final hyperparamter settings $\lambda$ for GNN.

| Dataset | Layers | GraphSAGE | GCN |
|---------|--------|-----------|-----|
| PubMed | 2 | 0.3909 | 0.0367 |
|  | 3 | 0.1436 | 0.1994 |
|  | 4 | 0.5485 | 0.8831 |
| Amazon-Photo | 2 | 0.0012 | 0.0069 |
|  | 3 | 0.3485 | 0.0855 |
|  | 4 | 0.0001 | 0.0024 |
| Amazon-Computers | 2 | 0.0014 | 0.0025 |
|  | 3 | 0.0037 | 0.0008 |
|  | 4 | 0.0021 | 0.0028 |
| WikiCS | 2 | 0.0274 | 0.0572 |
|  | 3 | 0.0142 | 0.0669 |
|  | 4 | 0.2226 | 0.0015 |

## C.3 Hyper-parameter Settings

We have only one unique hyperparameter, i.e., $\lambda$ in our methods, and the detailed settings are given in the Tab. 2 and Tab. 3. The search methods and search range are given in the main text. All the other hyper-parameters are set the same as baselines.

## C.4 Details of Datasets

In this section, we describe detailed information about the public available benchmark data sets used in our experiments.

**MNIST** [6] is a handwritten digits dataset containing 60000 training images and 10000 testing images. It is used to examine the performances of MLP with different hidden layers. **CIFAR-10** [5] is used for ResNet-18 and VGG-19, which consists of 50,000 $32 \times 32$ training color images and 10,000 testing images categorized as 10 classes. **ImageNet** is a benchmark dataset used for ResNet-50, which contains 14,197,122 annotated images with 1000 classes. For GNN, we selected four real-world graph datasets: **PubMed** [11] is a paper citation network where nodes represent documents and edges represent citation links. Both **Amazon-Photo** and **Amazon-Computers** [9] are subgraphs of Amazon co-purchase graph where nodes correspond to goods and edges connecting two nodes denote they are frequently bought together. **WikiCS** [8] is derived from Wikipedia whose nodes correspond to computer science articles and edges are hyperlinks. The license and source of these datasets are shown in Tab. 4 and Tab. 5, respectively. All the datasets are public and widely used in many research works. The collectors of the datasets are responsible for ensuring no privacy and consent issues.

Table 4: The license of datasets used in this paper.

| Dataset | license |
|---------|---------|
| MNIST | Creative Commons Attribution-Share Alike 3.0 license |
| CIFAR-10 | MIT license |
| ImageNet | MIT license |
| PubMed | NLM license |
| WikiCS | MIT license |
| Amazon-Photo | MIT license |
| Amazon-Computers | MIT license |

# D Dynamics of GNN and CNN during training procedure

In this section, we study the training procedure of ResNet-18 (CNN) and GraphSAGE-2 (GNN) of the vanilla version and model with NSR. The cross entropy loss and the expectation of neuron

Table 5: The source of datasets used in this paper.

| Dataset | link |
|---|---|
| MNIST | http://yann.lecun.com/exdb/mnist/ |
| CIFAR-10 | https://www.cs.toronto.edu/ kriz/cifar.html |
| ImageNet | https://image-net.org/index |
| PubMed | https://github.com/shchur/gnn-benchmark/raw/master/data/planetoid/ |
| WikiCS | https://github.com/pmernyei/wiki-cs-dataset |
| Amazon-Photo | https://github.com/shchur/gnn-benchmark/raw/master/data/npz/ |
| Amazon-Computers | https://github.com/shchur/gnn-benchmark/raw/master/data/npz/ |

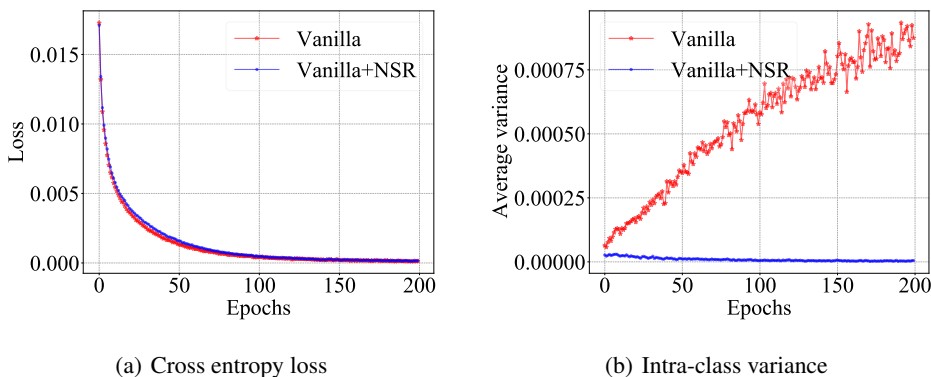

(a) Cross entropy loss

(b) Intra-class variance

Figure 1: Training procedure of vanilla ResNet-18 and Resnet-18 with NSR on CIFAR-10. (a) and (b) illustrate the corresponding cross entropy loss and average intra-class response variance of these two models on training set.

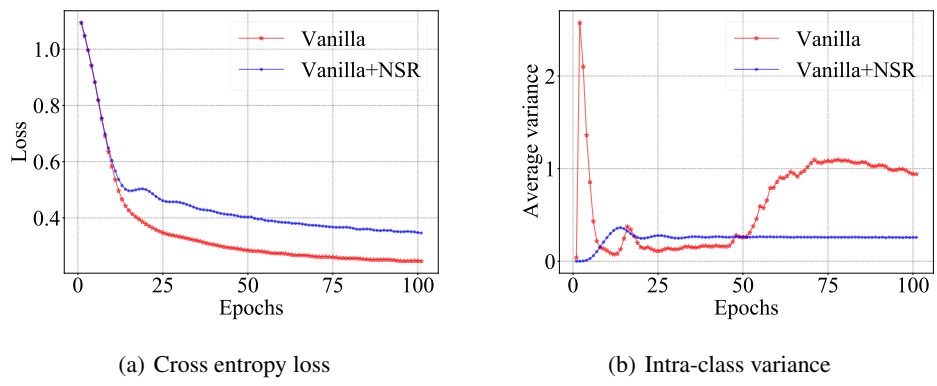

(a) Cross entropy loss

(b) Intra-class variance

Figure 2: Training procedure of vanilla GraphSAGE-2 and GraphSAGE-2 with NSR on PubMed. (a) and (b) illustrate corresponding cross entropy loss and average intra-class response variance of these two models on the training set.

intra-class response variance are demonstrated in Fig 1 and Fig 2. The results of GraphSAGE-2 and ResNet-18 show a similar tendency with MLP-4 discussed in Section 2.2. Both figures illustrate that the model with NSR shows smaller intra-class response variance which leads to high test accuracy shown in Sec 4.2.1. The overall increasing tendency of intra-class variance on GraphSAGE-2 is similar to the tendency of MLP as expected. However, there exist a few outliers. A potential reason is that the sampling process of GraphSAGE will introduce noise in the generated features which leads to perturbed responses of individual neurons during the early training phase.

## E  Time and space complexity analysis

In this section, we elaborate on details of the time and space consumption of NSR. The additional computation time overhead of adding NSR to the model is small. Tab. 6 shows the comparison of time consumption for one training epoch between the vanilla models and models with NSR respectively on MLP, ResNet-18 (CNN) and GraphSAGE-2 (GNN). We could see that the additional overhead is small especially for deeper networks like ResNet-18.

Table 6: Time (s) consumption of vanilla models and models with NSR on one training epoch.

| Model | MLP-4 | ResNet-18 | GraphSAGE-2 |
|---|---|---|---|
| Vanilla | 2.19 | 15.45 | 0.0489 |
| Vanilla+NSR | 2.54 | 16.56 | 0.0569 |

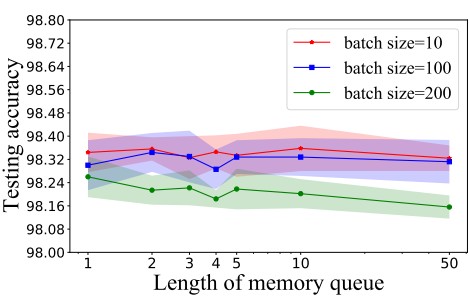

Figure 3: The testing accuracy of MLP-4 with NSR on MNIST with different memory queue length. The shadow indicates the standard deviation of the results obtained from five different runs with five different random seeds.

The additional space consumption is also small. The space complexity for NSR can be denoted as $O(N \times J \times M)$, where $N$ is the number of neurons in a layer, $J$ is the number of categories and $M$ is the length of memory queue. According to Fig. 3, the performance of MLP model is robust with the length of memory queue and a small memory queue length can still lead to a competing performance. So the length of memory queue can be set as a small constant. All experiments set the length of memory queue as 5. Therefore, the computation space complexity can be directly denoted as $O(N \times J)$. We show that this space complexity is more negligible than that of vanilla model, including all model weights and intermediate outputs of every layer. Using MLP as an example, the space complexity of weights of a layer can be represented as $O(N_{in} \times N)$, where $N_{in}$ represents the input feature dimension. The space complexity of output for a layer can be denoted as $O(N \times B)$, where $B$ denotes batch size. Since the number of categories $J$ is usually smaller than the number of neurons $N_{in}$ and batch size $B$, the space complexity of NSR is smaller than that of weights and output for a layer. The space used for NSR is usually much smaller than the space used for storing weights and layer output.