# OpenReview forum: "Neuron with Steady Response Leads to Better Generalization"
_NeurIPS.cc/2022/Conference — NeurIPS 2022 Accept_

### Official Review · Reviewer_CFBo · 2022-07-07

**Rating:** 6
**Confidence:** 3
**Soundness:** 3 good
**Presentation:** 3 good
**Contribution:** 2 fair

**Summary:**

This paper suggests that neurons with similar responses to instances of the same class may lead to better generalization. Accordingly, the authors propose a regularization method to reduce neuron intra-class response variance. This method outperforms the vanilla version of the optimization algorithm under the given settings on MLP, CNN, and GNN models. And the computational cost seems acceptable.

**Questions:**

- When labels of training data get corrupted, how will the performance of NSR be affected? Will the Consistency of Representations Complexity become less informative when the given labels are not the ground truth labels?
- When using selecting the layer to apply NSR, does the model need to be trained for a few epochs, and why?


**Limitations:**

Utilizing NSR method requires well-annotated data, which limits it to only supervised learning problems (though the reviewer don't consider this as a major flaw and there is no need to work on solving it here).

**Strengths And Weaknesses:**

Pros:
- The proposed regularization method is consistent with the idea of the Consistency of Representations Complexity and is well-motivated in this sense.
- The authors propose several practical ways to reduce computational budget and storage cost, e.g., saving a sequence of summation values of neurons' responses or regularizing only one particular layer. The experimental results show the effectiveness of these adjustments.
- This paper is well organized and clearly written. Most claims are well supported by theoretical analysis or experimental results.

Cons:
- The experiments are somehow limited. The generalization performance of models when using NSR under input corruptions is unclear. Given that this method requires the label information, evaluating the robustness to label noise is kind of necessary to verify its practical values.
- In order to utilize this regularization efficiently, one needs to choose the layer with the largest aggregated neuron intra-class response variance, which seems not to be directly accessible and requires additional calculations.

---

> ### Author Response · Authors · 2022-08-02
> **Thanks for your great questions and we conduct experiments to explore the robustness of models with NSR.**
>
> Thank you for your valuable comments and feedback. Here are our answers and some improvements based on your comments.
>
> 1. **Question about the robustness of models applied with NSR if input label has noise.**
> **Answer:** We follow the setting of this work[1] and conduct an experiment where the label of each image in CIFAR10 is randomly corrupted with a probability of 5\%. The results show that the error rate of vanilla ResNet18 increases from 4.22\% to 5.84\%, and vanilla ResNet18 with NSR can obtain an error rate of 5.25\% (10.1\% relative improvement). This result can show a certain degree of robustness of models with NSR.
>
> 2. **Comment that "one needs to choose the layer with the largest aggregated neuron intra-class response variance, which seems not to be directly accessible and requires additional calculations"**
> **Answer:** Yes, it needs some additional calculations. However, according to our experience, the layer with the largest aggregated neuron intra-class response variance could usually be estimated by running just a few updates (instead of epochs). So the additional calculation is usually quite small. We will add more explanations to the appendix. By the way, it is worth mentioning that even if NSR is applied on the other layer instead of the one with the largest aggregated intra-class response variance, the model will still achieve a good performance gain (please refer to Table 6 \& 7) as applying NSR on different layers have overlap effects.
>
> Thank you again for your constructive reviews. Hope that our response and additional experiment results can address your concerns. We will feel grateful if you could boost our paper.
>
> [1] Chiyuan Zhang, Samy Bengio, Moritz Hardt, Benjamin Recht, Oriol Vinyals. Understanding deep learning requires rethinking generalization. International Conference on Learning Representations (ICLR), 2017.

---

> > ### Comment · Reviewer_CFBo · 2022-08-06
> > **Reviewer Comment**
> >
> > Thanks for the answers and additional experiments. Overall I think the idea of NSR is interesting and insightful, and I've raised my score accordingly.

---

> > > ### Author Response · Authors · 2022-08-09
> > > **Thank you for your appreciation.**
> > >
> > > We sincerely thank you for your hard work in the reviewing period of NeurIPS'22 and we feel grateful for your helpful suggestions and questions. We are also very willing to further discuss with you if you have any questions or suggestions.

---

### Official Review · Reviewer_4vNZ · 2022-07-10

**Rating:** 6
**Confidence:** 4
**Soundness:** 3 good
**Presentation:** 2 fair
**Contribution:** 3 good

**Summary:**

This paper studies the intra-class neuron response variance in neural networks. The authors observe that this variance is lower for correctly classified samples, and hence propose to use this as a regularization term. They then show the effectiveness of this regularization compared to the vanilla case and other regularizations such as the Jacobian norm.

**Questions:**

1- It is claimed that neuron intra-class response variance has a high correlation with classification accuracy. However, this has not been directly shown directly. It is only shown that the variance is lower for correctly classified samples. How would a plot look up with the y-axis as accuracy and the x-axis as the variance for various experimental settings? This should be then compared with similar metrics in terms of correlation such as output sensitivity to input perturbations which are studied in papers like [1]. Then it would be interesting to see the particular information this metric brings as opposed to the previous ones.

2- Why not consider the zero response in the calculations? Even an inactivated state contains information by itself.

3- Have you studied the effect of using different regularization intensity values (lambda_n) for each sample? What is the effect of that?
If this has not been studied, then I would suggest not introducing it at all and only presenting a single lambda.

4- For MLPs deeper models observe a higher performance gain. However, this is not observed for CNNs (in Table 2). What could be the explanation for this?


[1] Sensitivity and Generalization in Neural Networks: An Empirical Study, Novak et al., ICLR 2018.

**Limitations:**

The mention "We also systematically consider the border impact, No risk is found."
They don't mention the limitations of the method and where it would possibly break.

**Strengths And Weaknesses:**

Strengths:

1- There are enough experiments presented to show the improvement compared to the vanilla case. It would be even better to have the same amount of experiments to compare with other regularization techniques as well (they are only presented in Table 4).

2- The metric is novel and interesting to the community.

Weaknesses:

1- The paper structure needs improvements. Section 2 is repetitions of what was already presented in Section 1. Also, the paper needs proofreading to fix typos.

2- Most of the ablation studies (comparing deeper and shallower networks and layer selection for computation) are done on MLPs. Performing these studies on CNNs is more interesting to practitioners.

3- Comparison of the neuron response variance to similar metrics such as input-output sensitivity is missing.

---

> ### Author Response · Authors · 2022-08-02
> **Thanks for the valuable questions and we add more explanations.**
>
> Thank you for your thoughtful feedback. Here are our answers and some improvements based on your comments.
>
> 1. **Comment that ''Section 2 has some repetitions of what was already presented in Section 1.''**
> **Answer:** Thank you for your comment. The reason for such repetitions is that we would like to highlight the correlation between neuron intra-class response variance and the classification correctness, so a few sentences in Section 1 and Section 2 may present similar points. Besides, such highlight in Section 2 is the foundation of the following designs and makes the paper more coherent. If you have more detailed suggestions for the writing, we would be glad to adopt them in the revision.
>
>
>
> 2. **Question about "Comparison of the neuron response variance to similar metrics such as input-output sensitivity".**
> **Answer:** Thanks for providing the detailed survey paper about the sensitivity. As described in the paper, the sensitivity can be measured by the norm of the input-output Jacobian. In addition, we have compared our NSR with the Jacobian-based regularization method (see Table 4), which shows that our NSR has a much lower error rate.
>
> 3. **Suggestion that ''plotting the figure with the y-axis as accuracy and the x-axis as the variance for various experimental settings to show the correlation between intra-class response variance and classification accuracy.''**
> **Answer:** Thanks for providing the optional way besides Figure 1 to show the correlation between intra-class response variance and classification accuracy. Under the constraint of space, we carefully compare Figure 1 and your suggested figure and choose the former in the revision as it may be easier to explain and understand for general readers. But we will add the new figure according to your suggestion in the appendix.
>
> 4. **Question on ''Why not consider the zero response in the calculations".**
> **Answer:** Thank you for your great question. In fact, we tried both strategies (i.e., consider or not consider the zero response in the calculations) and found that ignoring
> zero during intra-class response variance has better performance. The intuition is that zero response means the neuron is inactive (i.e., no response) which is irrelevant to presenting how stable the neuron response is. We can add this discussion in the appendix.
>
> 5. **Question on "the effect of using different regularization intensity values ($\lambda_n$) for each sample"**
> **Answer:** Thank you for your question. It is worth mentioning that $\lambda_n$ is the same for different samples. Actually, the subscript $n$ refers to neurons instead of samples, and this explanation can be found in line 122.
>
>
> 6. **Question on "For MLPs deeper models observe a higher performance gain. However, this is not observed for CNNs (in Table 2). What could be the explanation for this"**
> **Answer:** Thank you for this question. In fact, the trend in CNN is similar to the trend in MLP. From Table 2 we can see that the gain of NSR on VGG-19 (11.97\%) is indeed higher than its gain on ResNet-18 (11.37\%), which shows that deeper CNN networks can achieve a better gain in performance with NSR. It is worth mentioning that ResNet-50 in Table 2 is for ImageNet, while ResNet-18 and VGG-19 are for CIFAR10. So, the performance gain on ResNet-50 is not directly comparable with the gain on ResNet-18 or VGG-19.
>
>
> Thank you again for your constructive reviews. Hope that our response can address your concerns. We will feel grateful if you could boost our paper.

---

> > ### Comment · Reviewer_4vNZ · 2022-08-08
> > **Thanks for the response**
> >
> > Thanks for the response and clarifications.
> >
> > I still think that a comparison with sensitivity or Jacobian is missing in terms of correlation to the classification accuracy. Also, what the suggested plot would bring in addition to Figure 1 is how well the correlation holds between the metric and the classification accuracy when comparing different models/settings.

---

> > > ### Author Response · Authors · 2022-08-09
> > > **Thanks for your response and we provide more explanations about comparisons with Jacobian or sensitivity.**
> > >
> > > Thank you for your response.
> > >
> > > We compare the sensitivity proposed in [1] and the Jacobian regularization in [2] (which is used in Table 4 for comparisons between classical regularization methods). We find that they are both based on the F-norm of the Jacobian matrix between input-output. It is also mentioned in [2] that their proposed Jacobian regularization is "in line with the observed correlation between the input-output Jacobian and generalization performance [1]"(in Section 3.1 of [2]).
> > > Considering the relation between the Jacobian regularization and the sensitivity metric proposed in [1], we think that our comparison with the Jacobian regularization could provide the desirable answer to your interested question about the comparison with the sensitivity metric, and the results show our superiority. Later, we will include [1] in the Related Work session and may conduct the exact comparison with the sensitivity metric proposed in [1] as you suggested.
> > >
> > > On the other hand, we agree with you that the plot of variance over accuracy can provide additional evidence for our observation. We will carefully consider how to conduct different experiment settings to obtain the wanted figure and add it to the appendix.
> > >
> > > [1] Sensitivity and Generalization in Neural Networks: An Empirical Study, Novak et al., ICLR 2018.
> > >
> > > [2] Hoffman, Judy, Daniel A. Roberts, and Sho Yaida. Robust learning with jacobian regularization. arXiv preprint arXiv:1908.02729 (2019).

---

### Official Review · Reviewer_WDqK · 2022-07-13

**Rating:** 7
**Confidence:** 4
**Soundness:** 3 good
**Presentation:** 4 excellent
**Contribution:** 3 good

**Summary:**

This paper proposes a novel regularization scheme called Neuron Steadiness Regularization (NSR) which aims to reduce variance between activations belonging to a class. NSR is well motivated and well situated among related work in the paper. It is further verified empirically tested on wide variety of benchmark/model-combinations.

**Questions:**

In addition to what I mentioned in Strengths And Weaknesses section, I have following suggestions/questions:

- I wonder, despite adding NSR to multiple layers have overlapping benefits, what happens if we employ it in *all* the layers? This will likely simplify usage of this method since now it's not required to explore which layer activations to use.
- Another suggestion I had was to study qualitative effects of training with NSR, for example, does NSR implicitly penalizes sharpness? If yes, since SAM (https://arxiv.org/abs/2010.01412) has a larger computational overhead, can NSR be used in lieu of it with similar benefits?
- l311 typo: "Actiavtion"

**Limitations:**

Yes

**Strengths And Weaknesses:**

Strengths
- The paper does a good job of clearly explaining the proposed method and even discusses practical concerns like estimation error accrued by statistics being used.
- Research questions and experiments are thorough and well thought out, especially how NSR fares against other regularization methods seems quite pertinent.

Weaknesses
- My main set of suggestions are related to making the paper stronger by how far we can battle test NSR in practical settings, more specifically
  - The improvements with NSR can be magnified using stronger baselines, for instance, this paper (https://arxiv.org/abs/2010.01412) obtained much lower Top-5 error rate when training ResNet-50 on Imagenet for 200 epochs.
  - In Table 4 and 5, one way to merge these would be to add additional results for maybe at least ResNet-18 where both dropout and BN are typically used. I would also try to combine positive regularization methods for all 3 models with NSR and see if they provide orthogonal improvements to NSR.

---

> ### Author Response · Authors · 2022-08-02
> **Thanks for your valuable comments and we achieve some preliminary results.**
>
> Thank you so much for your great suggestions. We have added several improvements according to your suggestions.
>
> 1. **Suggestion that ''the improvements with NSR can be magnified using stronger baselines, such as the one used in SAM (SAM refers to the method in https://arxiv.org/abs/2010.01412) for ImageNet.''**
> **Answer:** Thank you for your great suggestion. We conducted an experiment based on the SAM (the mentioned paper). After 100 training epochs, we obtain the top-5 error rates: 6.96\% (w/o our regularization NSR), and 6.31\% (with NSR). NSR still achieves a **relative improvement of 9.33\% without tuning hyper-parameters**. Note that the ResNet-50 in SAM is trained by **TPU** with huge memory so that the batch size can be set to an extremely large value, for example, 4096 in their work. Nevertheless, we can only train their model on **GPU** and the largest batch size we can reach is only 800, 5 times smaller than theirs. Other hyper-parameters may need to be tuned accordingly; however, due to such computational resource constraints and the limited time, the result of our reproduced SAM is 6.96\%, instead of 6.28\% reported in the original paper.
>
>
> 2. **Suggestion that ''in Table 4 and 5, one way to merge these would be to add additional results for maybe at least ResNet-18 where both dropout and BN are typically used and to combine positive regularization methods for all 3 models with NSR and see if they provide orthogonal improvements to NSR.''**
> **Answer**: Thank you for your suggestion. The description of our experiment settings may be unclear, and we feel sorry for the potential misunderstandings. Actually, following common empirical settings in CV, the vanilla CNN models in Table 2 are trained with most of the widely-used regularization methods, including BN, dropout, learning rate decay, L2 norm and data augmentation. Some of them are mentioned in line 247. As a result, ''Vanilla + NSR'' is designed to verify the effect of combining NSR with the aforementioned regularization. Experiment results in Table 2 demonstrate that NSR can indeed provide orthogonal improvements beyond such methods. In Table 4, we remove most of the regularization methods to compare our proposed NSR with conventional regularization methods fairly. We have added more explanation in our revised manuscript (colored blue in line 255).
>
> 3. **Question that ''despite adding NSR to multiple layers has overlapping benefits, what happens if we employ it in all the layers?''**
> **Answer:** If we employ NSR to all the layers, the additional gain will be relatively little compared with NSR on a single layer, but the memory costs will be extremely high for the design of the memory queue. Thus, we only apply the regularization term on one layer to balance the effectiveness with the memory costs.
>
> 4. **Question that ''another suggestion I had was to study qualitative effects of training with NSR, for example, does NSR implicitly penalize sharpness? If yes, since SAM (https://arxiv.org/abs/2010.01412) has a larger computational overhead, can NSR be used in lieu of it with similar benefits?''**
> **Answer:** Thank you for pointing out a new perspective of sharpness. Currently, we have not conducted a theoretical analysis of NSR from this perspective, but we can provide an intuitive discussion. The underlying principle of NSR is different from sharpness. Sharpness indicates that the small change in model parameters will lead to a large change in loss function value. So, it regards the relation between model parameter change and the loss change. While NSR considers the stableness of intra-class response, which regards the relation between response change and input change. Besides, according to the empirical results, we believe NSR can provide some orthogonal improvements apart from penalizing sharpness. As mentioned in the first answer, we conduct experiments based on SAM, and applying our NSR obtains additional gains. We add the discussion of SAM in Related Work (which is colored blue in line 306) and will  explore this direction in future work.
>
> 5. l311 typo: "Actiavtion"
> **Answer:** Thank you for pointing it out. We have revised it in the new version of our manuscript (which is colored blue in line 311).
>
> Thank you again for your constructive reviews. Hope that our response and additional experiment results can address your concerns. We will feel grateful if you could boost our paper.

---

> > ### Comment · Reviewer_WDqK · 2022-08-09
> > **Response to the authors**
> >
> > Thanks to the authors for their changes!
> >
> > I've raised my score and I hope the authors generalize their insights and continue to explore this interesting direction, especially from an understanding standpoint. One more interesting experiment perhaps is to see if increasing weight decay just for the last layer classifier weights can approximate reducing intra class variance. [1] observed good results with this.
> >
> > [1] https://arxiv.org/abs/2106.04560

---

> > > ### Author Response · Authors · 2022-08-09
> > > **Thank you for your response.**
> > >
> > > We would like to thank you for your efforts in reviewing our paper and rebuttal materials. We really appreciate your detailed comments and valuable suggestions. We will conduct further experiments based on your suggestion to investigate the effect of weight decay on controlling intra-class variance. If you have any additional considerations, please let us know to make revisions accordingly.

---

### Public Comment · ~Yechao_Zhang1 · 2023-01-29
**Highly resembles the neural collapse phenomenon.**

The Neuron Steadiness Hypothesis raised in this paper resembles the neural collapse (NC) phenomenon in [1].

> **Neuron Steadiness Hypothesis**: neurons with similar responses to instances of the same class, i.e., more minor neuron intra-class response variance can lead to better generalization.

> **NC1**: Cross-example within-class variability of last-layer training activations collapses to zero, as the individual activations collapse to their class means.

NC happens at *the terminal phase of training (TPT), which begins at the epoch where training error ﬁrst vanishes*, and NC has substantiated *important beneﬁts, including better generalization performance, better robustness, and better interpretability*.

[1] Papyan V, Han X Y, Donoho D L. Prevalence of neural collapse during the terminal phase of deep learning training[J]. Proceedings of the National Academy of Sciences, 2020, 117(40): 24652-24663.

---

### Meta-Review · Area_Chair_thgc · 2022-08-28

**Recommendation:** Accept
**Confidence:** Less certain

**Metareview:**

This paper measures intra-class neuron response variance, and shows that network performance is better when it is lower. They then use this term as a regularization target, and show that it leads to improved model performance.

Reviews were high quality. Scores were between weak accept and accept, with one reviewer raising their score from weak accept to accept. The most significant concerns were experimental: around ablations, around the diversity and scale of models the technique was tested on, and around the tuning of baselines. However, the experiments seemed fairly strong as is, and of course there are always more experimental conditions that can be requested.

Based upon the reviewer consensus, I also recommend acceptance for this paper.

**Award:**

No

---

### Decision · Program_Chairs · 2022-09-14

Accept